# Direct penetration of spin-triplet superconductivity into a ferromagnet in Au/SrRuO$_3$/Sr$_2$RuO$_4$ junctions

M.S. Anwar[1], S.R. Lee[2,3], R. Ishiguro[4,5,6], Y. Sugimoto[1], Y. Tano[4], S.J. Kang[2,3], Y.J. Shin[2,3], S. Yonezawa[1], D. Manske[7], H. Takayanagi[4], T.W. Noh[2,3] & Y. Maeno[1]

Efforts have been ongoing to establish superconducting spintronics utilizing ferromagnet/superconductor heterostructures. Previously reported devices are based on spin-singlet superconductors (SSCs), where the spin degree of freedom is lost. Spin-polarized supercurrent induction in ferromagnetic metals (FMs) is achieved even with SSCs, but only with the aid of interfacial complex magnetic structures, which severely affect information imprinted to the electron spin. Use of spin-triplet superconductors (TSCs) with spin-polarizable Cooper pairs potentially overcomes this difficulty and further leads to novel functionalities. Here, we report spin-triplet superconductivity induction into a FM SrRuO$_3$ from a leading TSC candidate Sr$_2$RuO$_4$, by fabricating microscopic devices using an epitaxial SrRuO$_3$/Sr$_2$RuO$_4$ hybrid. The differential conductance, exhibiting Andreev-reflection features with multiple energy scales up to around half tesla, indicates the penetration of superconductivity over a considerable distance of 15 nm across the SrRuO$_3$ layer without help of interfacial complex magnetism. This demonstrates potential utility of FM/TSC devices for superspintronics.

[1] Department of Physics, Graduate School of Science, Kyoto University, Kyoto 606-8502, Japan. [2] Center for Correlated Electron Systems, Institute for Basic Science (IBS), Seoul 151-742, Republic of Korea. [3] Department of Physics and Astronomy, Seoul National University (SNU), Seoul 151-742, Republic of Korea. [4] Department of Applied Physics, Tokyo University of Science, Tokyo 162-8601, Japan. [5] RIKEN Center for Emergent Matter Science, Saitama 351-0198, Japan. [6] Department of Mathematical and Physical Sciences, Faculty of Science, Japan Women's University, Tokyo 112-8681, Japan. [7] Max-Planck-Institut fur Festkorperforschung, Stuttgart D-70569, Germany. Correspondence and requests for materials should be addressed to M.S.A. (email: anwar@scphys.kyoto-u.ac.jp).

Superconducting Cooper pairs are known to penetrate a normal metal (N) adjacent to a superconductor. This phenomenon, called the superconducting proximity effect, can occur even in FMs, where the strong ferromagnetic exchange field is expected to destroy the Cooper pairs. For FM/SSC interfaces, the proximity effect arises with the induction of spin-singlet Cooper pairs (pairs with the total spin $S = 0$) into the FM over a coherence length $\xi_F = \sqrt{\frac{\hbar D_F}{E_{ex}}}$, where $\hbar$, $D_F$, $E_{ex}$ are the reduced Planck constant, diffusion coefficient and exchange energy, respectively. Importantly, $\xi_F$ is no more than a few nanometres even for weak FMs[1]. However, spin-triplet Cooper pairs (pairs with $S = 1$) can also emerge at a FM/SSC interface if appropriate magnetic inhomogeneity, such as the ferromagnetic domain walls or noncollinear magnetization, exists at the interface[2,3]. Since spin-triplet Cooper pairs with parallel spins ($S_z = \pm 1$) are insensitive to exchange field, they can penetrate the FM over a distance much longer than $\xi_F$. This is called the 'long-range proximity effect' (LRPE) and it has actively been investigated[3–8]. However, to exploit the SSC-based LRPE in spintronics devices, the necessity for magnetic inhomogeneity can be a technical issue, since magnetic inhomogeneity itself complicates the fabrication and can even disturb the spin information.

In contrast, there is a straightforward means of inducing the LRPE. This involves using TSCs instead of SSCs. Favourably, even without magnetic inhomogeneity, both charge and spin supercurrents can be generated at the FM/TSC interface[9–12]. The essential simplicity and the utility of the spin-degree of freedom in a FM/TSC junction in comparison with a FM/SSC junction is depicted in Fig. 1a,b. The absence of magnetic inhomogeneity is advantageous to transfer spin information with less disturbance. Furthermore, by rotating the magnetization of the FM relative to the spin of the spin-triplet Cooper pairs by applying an external magnetic field, the induction and functionality of the LRPE can be controlled. To date, the proximity effects at a FM/TSC interface have not been realized experimentally.

In case of FM/TSC junctions, the orbital symmetry of the induced superconducting correlation is also important. In any superconductor-based junctions, because of the lack of inversion symmetry near the interface, pair amplitudes

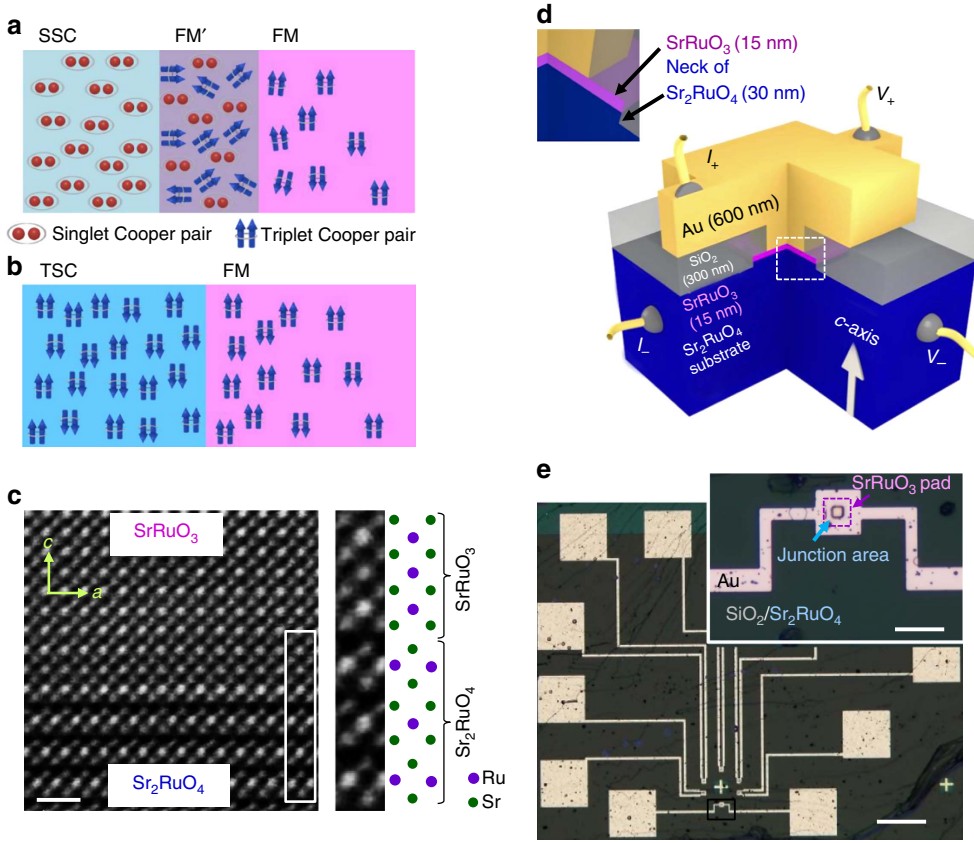

**Figure 1 | Structure of Au/SrRuO₃/Sr₂RuO₄ devices.** (**a**) A schematic depiction of a spin-singlet superconductor (SSC) based ferromagnetic-metal (FM)/FM′/SSC junction, where a special kind of complex magnetic inhomogeneity (the FM′ layer) is always required. (**b**) For spin-triplet superconductor (TSC) based FM/TSC junctions, spin-polarized supercurrent directly penetrates into FM without any magnetic inhomogeneity, and importantly, with undisturbed spin information. (**c**) Transmission electron microscope image of the interfacial region of a 15-nm-thick SrRuO₃ thin film deposited on the cleaved *ab* surface of Sr₂RuO₄ substrate taken along its [010] direction. The scale bar at the left bottom corresponds to 1 nm. The region indicated by the white rectangle is shown at a higher magnification on the right-hand side, together with a schematic illustration of the Ru (purple) and Sr (green) atomic positions. (**d**) Three-dimensional (3D) schematic image of a Au(600-nm)/SrRuO₃(15-nm)/Sr₂RuO₄ junction. SrRuO₃ square pads of the area 25 × 25 μm² and 10 × 10 μm² are formed by etching to fabricate the 20 × 20 μm² (Junction A) and 5 × 5 μm² (Junction B) junctions. A 300-nm thick SiO₂ layer is deposited to electrically isolate the top Au electrode from the Sr₂RuO₄ substrate. As shown in the inset, during the fabrication process, the Sr₂RuO₄ substrate surface was etched by ≈30 nm except for the area under the SrRuO₃ pads, resulting in 'necks' of Sr₂RuO₄. (**e**) Optical microscope image of a device containing six different junctions. The brighter regions are the top Au electrodes. The scale bar at the right bottom corresponds to 200 μm. Inset shows a magnified image taken within the black rectangular area in which a 5 × 5 μm² junction (step in the center of Au pad) is located. The purple square indicates the SrRuO₃ pad (10 × 10 μm²) under the top Au electrode. The scale bar is equivalent to 20 μm.

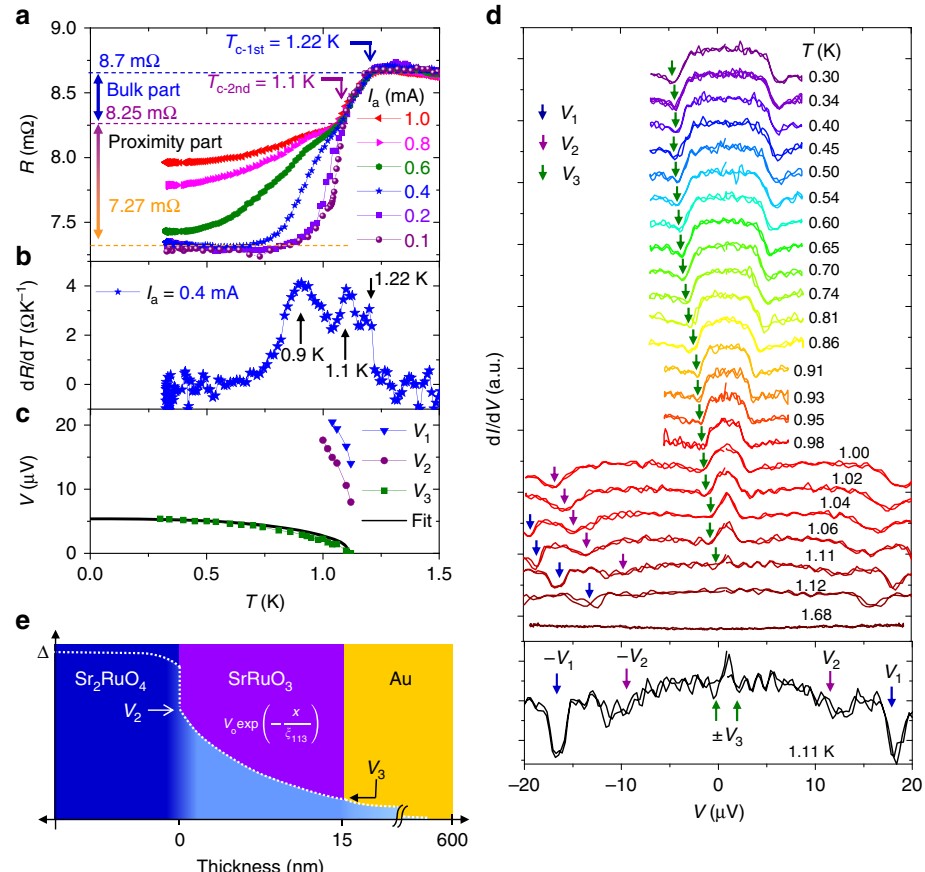

**Figure 2 | Temperature-dependent transport properties of a junction. (a)** Resistance ($R$) of Junction A as a function of temperature ($T$) within a temperature range of 1.5–0.3 K, measured at different applied currents ($I_a$) between 0.1 and 1 mA. Two obvious superconducting transitions are observed. The first transition ($T_{c\text{-1st}}$), at 1.22 K, corresponds to the bulk superconductivity of $Sr_2RuO_4$ at the neck below the $SrRuO_3$ pad. The second transition ($T_{c\text{-2nd}}$) at 1.1 K arises at the $SrRuO_3$/$Sr_2RuO_4$ interface due to the proximity effect. **(b)** Temperature derivative of resistance ($dR/dT$) measured at $I_a = 0.4$ mA. Three clear peaks indicate three transitions in the junctions. **(c)** Temperature-dependent characteristic voltages at zero field: $V_1$ (blue triangle), $V_2$ (purple circle) and $V_3$ (green squares). The solid line is a fit with a temperature dependent $p$-wave superconducting gap of $Sr_2RuO_4$. **(d)** Differential conductance ($dI/dV$) versus applied voltage for temperatures between 1.12 and 0.3 K measured at zero field after zero field cooling. The curves are shifted vertically for clarity. The lower panel shows a $dI/dV$ curve at 1.11 K to emphasize the three features. This curve exhibits a clear anomaly at a characteristic voltage $V_1 \approx 14.5\,\mu V$ (blue arrows) and a slight enhancement in the conductance appears at $V_2 \approx 8.5\,\mu V$ (purple arrows). Furthermore, an additional enhancement emerges within $\pm V_3$ (green arrows). **(e)** Schematic depth profile of superconducting order parameter (dashed white line) induced in the Au/$SrRuO_3$/$Sr_2RuO_4$ junction. $V_2$ and $V_3$ correspond to the order parameters at the interface between $SrRuO_3$/$Sr_2RuO_4$ and Au/$SrRuO_3$, respectively.

with various orbital symmetries ($s$-wave, $p$-wave, $d$-wave and $f$-wave) can in principle emerge at the interface[13]. Regardless the nature of the bulk superconductor, for a diffusive and clean junction the $s$-wave odd frequency and $p$-wave even frequency Cooper pairs dominate, respectively. So far, experimentally, $s$-wave odd frequency correlation has been observed in Josephson junctions (in diffusive limit) fabricated using various ferromagnets and $s$-wave SSCs. On the other hand, the $p$-wave correlation has never been realized. Thus, FM/TSC junctions provide opportunity to study properties of the $p$-wave induced pair amplitude[14], in particular toward utilization of the orbital degree of freedom in combination with the spins.

$Sr_2RuO_4$ (SRO214) with the superconducting critical temperature ($T_c$) of 1.5 K is one of the best-candidates TSCs[15]. Extensive experimental and theoretical studies[16] indicate that SRO214 exhibits most likely the chiral $p$-wave spin-triplet state with broken time-reversal symmetry[17–22], although there are still unresolved issues[23,24]. Recently, SRO214 attracts interest for exploring topological superconducting phenomena originating from its orbital phase winding[16]. Investigations of

FM/SRO214 would also provide definitive evidence for its superconducting order parameter.

Since high-quality superconducting SRO214 thin films are not available, we recently developed an alternative method to obtain FM/SRO214 heterostructures. We use SRO214 single crystals as substrates and epitaxially deposit ferromagnetic $SrRuO_3$ (SRO113) thin films with an atomically smooth and highly conductive interface[25]. By utilizing SRO113(15-nm)/SRO214 heterostructures and *ex-situ* depositing 600-nm-thick Au layer as an N layer, we fabricate N/FM/TSC junctions (see Methods).

In this Letter, we present results demonstrating for the first time the emergence of the spin-triplet LRPE in a FM through a simple epitaxial interface. It is essentially different from previous FM/SSC junctions, for which magnetic inhomogeneity is always required (Fig. 1a). The voltage-dependent differential conductance ($dI/dV(V)$) exhibits multiple characteristic features, persisting up to as high as 0.5 T without affected by ferromagnetic domain rearrangement. This result indicates that the superconducting correlations are induced even in the N-layer beyond the 15-nm FM layer, evidencing LRPE.

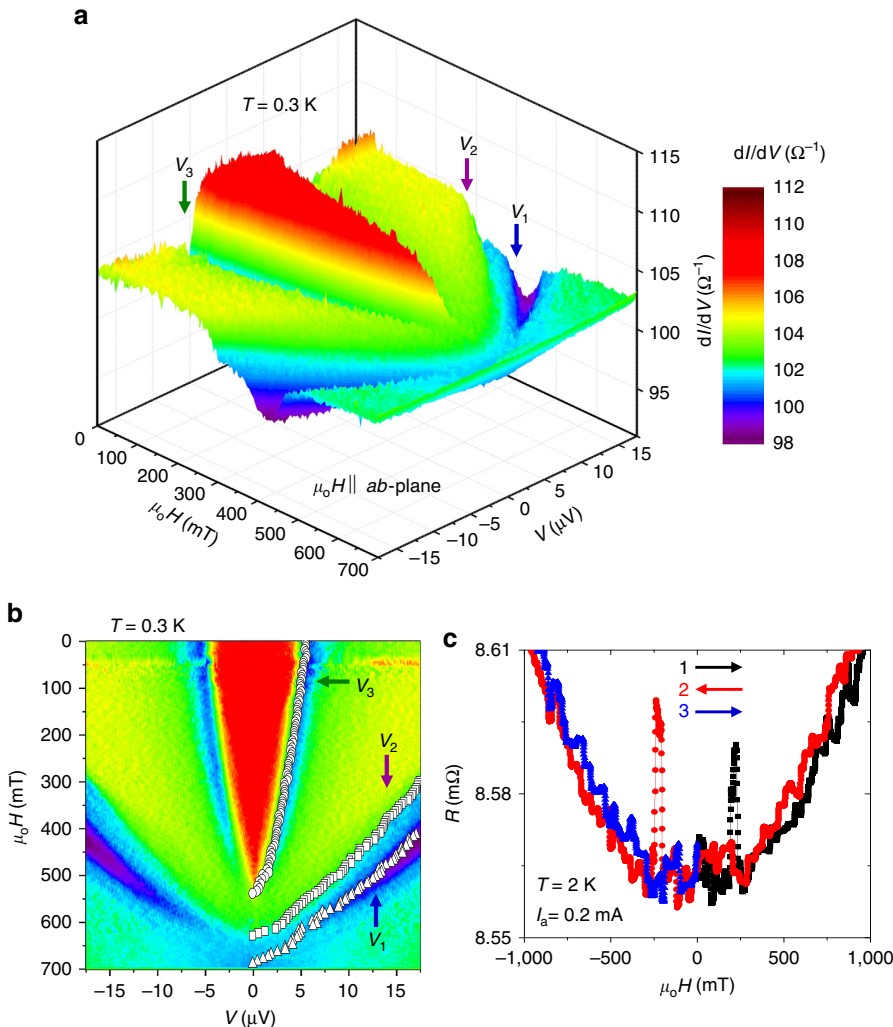

**Figure 3 | Magnetic-field dependent transport properties of a junction.** (**a**) 3D colour plot of differential conductance d$I$/d$V$ of Junction A as a function of the bias voltage ($V$) and applied field ($\mu_o H$) parallel to the interface from 0 to 700 mT measured at 0.3 K. Three features at voltages $V_1$ (blue arrow), $V_2$ (purple arrow) and $V_3$ (green arrow) are clearly seen. To generate this colour map, d$I$/d$V$ curves were obtained with field interval of 5 mT (Supplementary Fig. 5). (**b**) 2D colour contour map of d$I$/d$V$. On the right-hand side ($V > 0$), the values of $V_1$, $V_2$ and $V_3$ are also plotted. The colours are the same as those used in Fig. 3a. (**c**) Normal-state magnetoresistance measured at 2 K exhibiting two sharp hysteresis peaks around $\pm 200$ mT due to the enhanced scattering associated with the rearrangement of the ferromagnetic domains in the SrRuO$_3$ layer.

## Results

**Transport behaviour of a superconducting junction.** For an N/SSC interface with negligible interfacial electronic-transport barrier strength $Z$, the differential conductance $G_S$ of the interface is twice the normal-state conductance $G_N$ for a bias voltage within the superconducting-gap energy. This is due to the transmission of an electron as a Cooper pair with a reflected hole via the Andreev reflection[26]. As $Z$ increases, $G_S$ decreases within the gap and approaches zero for $Z \to \infty$. For an FM/SSC interface, $G_S$ falls with an increase in the spin-polarization of the FM and reaches zero for a 100% spin-polarized FM[27], even for $Z \approx 0$. Our Au/ SRO113/SRO214 junctions consist of two important interfaces: SRO113/SRO214 and Au/SRO113 (Fig. 1c–e). For such junctions, multiple features in the conductance are expected if the superconducting correlation reaches the Au/SRO113 interface as a result of LRPE.

**Temperature dependent transport properties.** Figure 2a shows the temperature dependent resistance of Junction A measured with different applied currents $I_a = 0.1$–$1.0$ mA. We observe the

decrease in the resistance (increase in conductance) due to superconductivity. Obviously, there are multiple superconducting transitions. The first transition occurs at 1.22 K, which corresponds to the bulk superconducting transition of the SRO214 neck part immediately below the SRO113 layer (Fig. 1d). Indeed, the drop in the resistance accompanying this first transition is robust against $I_a$ up to 1 mA. In addition, a resistance drop of 0.5 m$\Omega$ closely corresponds to the estimated value of the resistance of the SRO214 neck part (see Supplementary Figs 1 and 2 and Supplementary Note 1). The second transition emerges at 1.1 K for $I_a = 0.1$ mA. The superconducting feature of the second transition is significantly suppressed with increasing $I_a$. By taking the temperature derivative of the resistance measured at 0.4 mA, we clearly see in addition to two peaks corresponding to the first and second transitions, another broad peak at lower temperatures (Fig. 2b). The suppression of the second transition and the existence of the third peak in d$R$/d$T$ implies multiple characteristic energies induced within the junctions.

Next, we extract characteristic voltages associated with these three features. Figure 2c summarizes their temperature dependencies, as obtained from the d$I$/d$V(V)$ at different

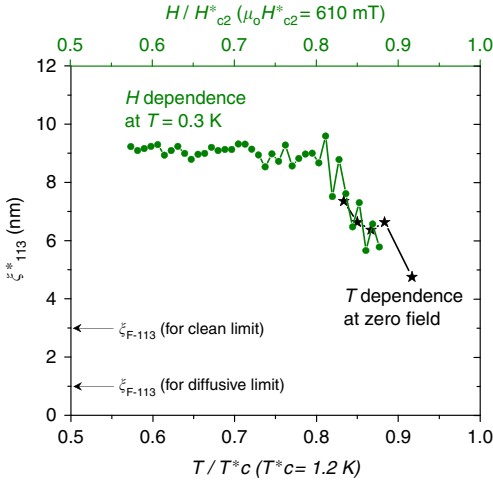

**Figure 4 | Characteristic decay length.** Superconducting correlation decay length in the SrRuO$_3$ layer $\xi_{113}^*$ as a function of the reduced temperature $T/T_c^*$ measured at zero field (black stars), and as a function of the reduced field $H/H_{c2}^*$ along the $ab$-surface at 0.3 K (green circles). For comparison, superconducting coherence length for spin-singlet correlation in SrRuO$_3$ ($\xi_{F-113}$) for diffusive (1 nm) and clean (3 nm) limits are indicated with arrows. We can obtain $\xi_{113}^*$ only in the temperature and field ranges where both voltages $V_2$ and $V_3$ appear. Note that, with decreasing temperature, $\xi_{113}^*$ increases and almost matches the field-dependent behaviour. The values of $T_c^*$ and $H_{c2}^*$ are determined based on the experimental data shown in Figs 2d and 3b.

temperatures (Fig. 2d). Note that we take the derivative of $I$–$V$ curves to obtain the d$I$/d$V$ curves (Supplementary Figs 3 and 4; Supplementary Note 2). At 1.12 K, immediately below the first transition, we observe strong dips at a characteristic voltage $V_1 \approx \pm 14\,\mu V$ (blue arrow) and a small conductance enhancement within $\pm V_1$. The corresponding current flow of $\approx 2\,mA$ yields current density of $\approx 10^7\,A\,m^{-2}$ through the SRO214 neck (Fig. 1b inset). Thus, the feature of $V_1$ originates from the destruction of the superconductivity at the neck by a current exceeding the critical current density of SRO214 (ref. 28). Interestingly, with decreasing temperature, another dip at $V_2$ (purple arrow) and the enhancement of the conductance within $\pm V_2$ become evident (Fig. 2d, lower panel). The appearance of $V_2$ provides a signature of the Andreev reflection with a small $Z$ at the SRO113/SRO214 interface, indicating that superconductivity is induced in SRO113 via the proximity effect. At 1.11 K, an additional enhancement starts to appear within $\pm V_3$. Although $V_3$ is close to zero at around 1.11 K, it increases to 5 μV at 0.3 K (Fig. 2d, green arrows). Because of the enhancement of the conductance with a flat-top peak at low temperatures, the $V_3$ anomaly is attributable to an additional Andreev reflection in the junction. The temperature dependence of characteristic voltages are shown in Fig. 2c. Interestingly, $V_3(T)$ is described well with the temperature dependence of the superconducting gap of Sr$_2$RuO$_4$ obtained by taking into account the $p$-wave paring and two dimensionality[29]. This fact supports our interpretation that $V_3$ represent superconducting-gap-like features.

We should comment here on the chiral $p$-wave $p_x + ip_y$ order parameter of SRO214. In some types of experiments, super-conducting contribution in transport along the $c$ axis should vanish because the $p_x + ip_y$ order parameter is averaged to zero. For example, in the $c$-axis oriented Josephson junctions between SRO214 and an $s$-wave superconductor, critical current is expected to be cancelled out to be zero. In contrast, the Andreev reflection in a metal attached along the $c$ axis is certainly allowed even for the $p_x + ip_y$ state because the magnitude of the Andreev reflection is not given by the simple integration of the order-parameter phase[14].

Let us consider where the Andreev reflection related to $V_3$ takes place. As mentioned above, $V_2$ should arise from the proximity induction at the SRO113/SRO214 interface. Considering the two interfaces within our junction, $V_3$ should arise at the other interface, that is, the Au/SRO113 interface. For this to occur, any superconducting correlations penetrating from the SRO113/SRO214 interface must reach the Au layer across the 15-nm SRO113 layer, as illustrated in Fig. 2e. Other possible explanations of these multiple features are discussed in Supplementary Note 3.

**Field dependent transport properties.** To investigate the magnetic field effect on these three features, we measured d$I$/d$V(V)$ at 0.3 K with the field applied along the $ab$-surface (in-plane), after zero-field cooling. The obtained data (Supplementary Fig. 5) are presented in Fig. 3a as a three-dimensional colour map, which clearly shows the three features. The field dependences of $V_1$ to $V_3$ plotted in Fig. 3b exhibit a smooth decrease as the field increases. In particular, $V_2$ and $V_3$ exhibit $(1 - \frac{H}{H_c})^{1/2}$ behaviour at high fields (Supplementary Fig. 6), as expected for gap-like features[30]. The critical fields $H_c$ for $V_2$ and $V_3$ reach as high as $\approx 550\,mT$ and $\approx 650\,mT$, respectively, indicating that the proximity effect is limited by the bulk superconducting gap but not by the interfacial magnetic structure. The smooth variations of $V_2$ and $V_3$ are also surprising because the ferromagnetic domain walls within the device change their configuration at $\approx 200\,mT$, as verified by the sharp peaks in the normal-state magnetoresistance at 2 K (Fig. 3c). These peaks are associated with enhanced domain-wall scattering. Thus, the domain walls do not play significant roles in the generation of the proximity effect.

## Discussion
Briefly summarizing the experimental observations, we observe three superconducting features $V_1$, $V_2$ and $V_3$ in d$I$/d$V(V)$. Conductance enhancement within $\pm V_3$ strongly suggests the penetration of superconducting correlations through the 15-nm SRO113 layer leading to the Andreev reflection at the Au/SRO113 interface. This penetration is found to persist even above 0.5 T and to be unaffected by the ferromagnetic domain-wall configuration of the SRO113 layer.

The observed superconducting proximity penetration of 15 nm in SRO113 is actually much stronger than the expected penetration for spin-singlet Cooper pairs. The ferromagnetic coherence length for SRO113 ($\xi_{F-113}$) of the spin-singlet proximity induction is 3 nm for the clean limit (1 nm for the diffusive limit)[31], which is five (fifteen) times smaller than the observed value. Therefore, only spin-triplet Cooper pairs can be responsible for the superconducting penetration within SRO113 of the present devices. Thus, this observation marks a new type of LRPE. Similar results are obtained from another junction (Junction B: $5 \times 5\,\mu m^2$; Supplementary Fig. 7). For comparison, various obtained parameters are given in Supplementary Table 1 (also see Supplementary Note 4).

The characteristic decay length of this new LRPE can be estimated by assuming that the superconducting gap decreases as $\Delta(x) = \Delta_o \exp(- \frac{x}{\xi_{113}^*})$, where $x$ is the distance in the SRO113 layer from the SRO214/SRO113 interface. From the relationship $V_2 \propto \Delta(x=0)$ and $V_3 \propto \Delta(x=15\,nm)$, we evaluate $\xi_{113}^* = 7\,nm$ at 1 K and zero field and $\xi_{113}^* = 9\,nm$ at 0.3 K and 350 mT. These values are about three times larger than $\xi_{F-113}$ for spin-singlet superconductivity in the clean limit but smaller than the thickness of the SRO113 layer. As shown in Fig. 4, we found that

$\xi^*_{113}$ increases as the temperature decrease; at 0.3 K, $\xi^*_{113}$ is field-independent between 350 and 500 mT, as would be expected for the LRPE.

This penetration of the spin-triplet pairs is naturally explained by assuming SRO214 to be a TSC, as already evidenced by many experiments[14,32]. On the other hand, it should be noted that spin-triplet pairs can be generated even from a SSC provided magnetic inhomogeneity exists at the FM/SSC interface[3–20]. The ferromagnetic domain walls also provide this magnetic inhomogeneity. In this case, we would expect the domain-wall configuration to severely affect junction properties; the alignment of the domains by external fields should strongly suppress the proximity effect. However, we observed that superconducting transport properties of our junctions are insensitive to the domain-wall configurations and thus LRPE in SRO113 should be induced without magnetic inhomogeneity. Therefore, a hypothetical scenario that SRO214 is an SSC and the LRPE is induced by magnetic inhomogeneity is unlikely. The present results constitute additional new evidence for spin-triplet superconductivity in SRO214. To support this scenario, we performed theoretical calculations explained in Supplementary Note 5 (also see Supplementary Fig. 8).

Generally, in addition to ordinary even-frequency spin-triplet $p$-wave Cooper pairs, odd-frequency spin-triplet $s$-wave pairs can be generated at the interface[9–11]. The former cannot survive over a distance greater than the electron mean free path $l_e$, whereas the latter remains stable over a greater distance. SRO113 is a relatively good metal at low temperatures and for thin films with residual resistivity of around 10–20 $\mu\Omega$ cm yielding, $l_e \approx 15$ to 30 nm (ref. 33). This is of the same order or greater than the thickness of the SRO113 layer used in our junctions. In such a case, the even-frequency $p$-wave correlation amplitude may dominate. Theoretically, it has been demonstrated that, for diffusive N/TSC junction, the odd-frequency $s$-wave correlation can emerge at the interface and lead to a sharp zero-bias peak in the conductance spectrum originating from the induced mid-gap Andreev resonant state[34]. We anticipate that a similar zero-bias peak would be observed also in FM/TSC junctions if the odd-frequency pairs were dominant. On the other hand, the conductance of the present junctions at lower temperatures and zero field (Fig. 2d) exhibits a flat-top peak around zero bias voltage, without any sharp anomalies. These observations also support the penetration of the $p$-wave (even-frequency) amplitude.

The LRPE can be also sensitive to spin-flip scattering events. Therefore, the Cooper pair penetration is also limited by the spin-flip length $L_{sf} = \sqrt{D_F \tau_{sf}}$, where $\tau_{sf}$ is the spin-flip time. Both $s$-wave (odd frequency) and $p$-wave (even frequency) correlation cannot extend over this length, whereas the latter can also be limited by the $l_e$. For SRO113 thin films, $\tau_{sf}$ is approximately $\approx 300$ fs as revealed by an optical Kerr effect experiment[35], yielding $L_{sf} \approx 30$ nm. This estimated $L_{sf}$ is three times larger than the value of the decay length $\xi^*_{113} \sim 10$ nm obtained from experiment. This comparison suggests that the proximity effect arising in our devices decays due to potential scatterings rather than spin-flip scatterings, in consistent with the scenario for the penetration of $p$-wave correlations.

Nevertheless, it is actually impossible to completely deny contributions of the odd-frequency $s$-wave pairs. To examine the orbital characteristics of the dominating Cooper pairs, further studies using junctions with different thickness of SRO113 and a tunnel barrier between N and F layers are needed.

In summary, spin-triplet Cooper-pair penetration occurs in the superconducting Au/SRO113/SRO214 junctions without additional inhomogeneous magnetic layer. The results are attributed to the direct penetration of even-frequency $p$-wave spin-triplet pairs from a TSC into a FM through a simple epitaxial interface and persist even at 0.5 T. This observation opens new possibilities to transfer $p$-wave superconductor across an interface without losing spin information. Thus, our research offers new directions for the utilization of LRPE in FM/TSC systems in a new research area termed Superspintronics.

## Methods

**SrRuO$_3$ thin films deposition.** High-quality SRO214 single crystals were grown by using a floating-zone method. SRO214 crystals with the intrinsic $T_c \approx 1.5$ K require the elimination of Ru deficiencies; as a result, some parts of the crystals tend to contain SRO327, SRO113, as well as Ru-metal inclusions. In this study, we carefully choose SRO214 crystals without any such inclusions, but with a slightly lower $T_c \approx 1.22$ K. Ferromagnetic SRO113 thin films were epitaxially deposited by means of pulsed laser deposition on the cleaved $ab$-surface of the SRO214 substrates ($3 \times 3 \times 0.5$ mm$^3$)[25]. Immediately after the deposition of the SRO113 thin film, a 20-nm thick Au layer with a 5-nm thick Ti adhesive layer was deposited $ex\text{-}situ$ as a capping layer by dc-sputtering. Transmission electron microscope image given in Fig. 1c shows that SrRuO$_3$/Sr$_2$RuO$_4$ interface is atomically smooth.

**Device fabrication.** Afterwards, laser ultraviolet maskless photolithography was utilized to fabricate the Au/SRO113/SRO214 junctions. First, micron-sized square pads of SRO113 of the area of $25 \times 25$ $\mu$m$^2$ (Junction A) and $10 \times 10$ $\mu$m$^2$ (Junction B) were etched out using a AZ1500 photoresist (baked at 110 °C for 2 min) and ion-beam etching (1 kV, 100 W). Next, a 300-nm SiO$_2$-layer was deposited (at a backing pressure of $10^{-9}$ mbar) over the samples, while maintaining $20 \times 20$ $\mu$m$^2$ and $5 \times 5$ $\mu$m$^2$ windows over $25 \times 25$ $\mu$m$^2$ and $10 \times 10$ $\mu$m$^2$ SRO113 pads, respectively, using a bilayer photoresist (AZ1500 and LOR15) and a lift-off technique. Note that the junction area is made smaller than the overall SRO113 pad area to avoid direct contact between the Au and SRO214. Top Au electrodes were also prepared by applying a lift-off technique. Finally, a 600-nm thick Au-layer was sputtered at a backing pressure of $10^{-9}$ mbar. Figure 1e shows optical microscope image of a device with six such junctions.

**Measurements of transport properties.** Electrical transport measurements were performed using the four-point technique with two contacts on the Au top electrode and the other two contacts connected directly to the side of the SRO214 crystal, as shown in Fig. 1d. Transport properties were studied down to 300 mK using a $^3$He cryostat equipped with a superconducting magnet. The temperature-dependent resistance in the normal state (300–4 K) of Au/SRO113/SRO214 junctions is dominated by the interfacial resistance, with additional contribution of the $c$-axis (out-of-plane) resistance behaviour of the bulk SRO214 substrate (Supplementary Fig. 2a). Furthermore, the normal-state magnetoresistance at 2 K exhibits two sharp peaks at 200 mT (Fig. 3d). These facts reveal that the applied current ($I_a$) properly passes through the interface.

**Data availability.** The data corresponding to this study are available from the corresponding author on request.

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

## Acknowledgements

We are grateful to Y. Tanaka, J. Aarts, S. Kashiwaya, Jian Wei and Ying Liu for the useful discussions. We are thankful to M. Cuoco, D. Terrade, P. Gentile and M. Kunieda for their help on theoretical calculations. M.S.A. is supported as an International Research Fellow of the Japan Society for the Promotion of Science (JSPS). This work was supported by Grants-in-Aid for Scientific Research on Innovative Areas on 'Topological Quantum Phenomena' (JSPS KAKENHI JP22103002) and 'Topological Materials Science' (JSPS KAKENHI JP15H05851 and JP15H05852); and by Grant-in-Aid JSPS KAKENHI JP26287078. This work was also supported by IBS-R009-D1 and by Kyoto University Research Development Program "Migaki" 2015.

## Author contributions

M.S.A. devised the concept, designed the experiments and prepared the SRO214 substrates from crystals grown in Y.M.'s group at Kyoto University. S.R.L. and Y.J.S. prepared the SRO113 thin films and S.J.K. examined the films using a transmission electron microscope, under the supervision of T.W.N. at Seoul National University. R.I., Y.T. and M.S.A. fabricated the devices under the supervision of H.T. at RIKEN and the Tokyo University of Science. M.S.A., Y.S. and S.Y. performed the transport measurements under the supervision of Y.M. at Kyoto. D.M. provided the theoretical input on the results. M.S.A., S.Y., Y.S. and Y.M. wrote the paper and all authors contributed in the discussions.

## Additional information

**Competing financial interests:** The authors declare no competing financial interests.

