## [Peer Review File · Nature Communications]

Reviewers' comments:

Reviewer #1 (Remarks to the Author):

Report on: "Direct penetration of spin-triplet superconductivity into a ferromagnet in Au/SrRuO₃/Sr₂RuO₄ junctions" by M. S. Anwar et al.

In their manuscript the authors present an experiment that show evidence of a long-range proximity effect of the superconducting state of Sr₂RuO₄ in the ferromagnetic SrRuO₃. This proximity effect is probed by a conductance measurement using a Au electrode on top the ferromagnet. Cooling the device down below 1.22 K two distinct reductions in resistance are seen and are correlated to: The superconducting transition in Sr₂RuO₄ at 1.22K as expected. The transition to a proximity superconductivity in the SrRuO₃ to layer. The SrRuO₃ layer is epitaxially grown on c-axis oriented Sr₂RuO₄ single crystal. This should give good electrical contact between the two materials.

The experimental findings in the low temperature regime of the conductance shows an emergence of an Andreev conductance at low bias. The bias window where the conductance is enhanced scales as the superconducting gap, either with temperature or with magnetic field, as expected from simple BCS theory. Furthermore, the conductance data shows a second structure found at higher biases with the same scaling as the low-bias feature. Combining the two features to a position-dependent proximity gap in SrRuO₃ a decay length, or effective coherence length, is extracted. This length exceeds estimates of a decay length expected in either a ballistic or diffuse ferromagnetic metal by factor between 3-10.

While I find the experiment very beautiful and the data extracted give a clear picture of a proximity effect extending over the 15nm SrRuO₃ ferromagnet, I find it hard to be convinced that this is an unequivocal evidence of Sr₂RuO₄ being a triplet superconductor. For this I would like to see some detailed modelling to support this claim. The chiral form of the pairing state in Sr₂RuO₄, the $(p_x + i p_y)$ orderparameter, would for instance average to zero in a c-axis oriented experiment as the current. How does this fact relate to the interpretation put forward? An other suggestion is that at the Sr₂RuO₄/SrRuO₃ interface there is a nucleation of a odd-frequency s-wave triplet pair amplitude. Such amplitude is always present where reflection symmetry is broken. However, what is the theoretical signal in the low-bias conductance of such a pair-amplitude?

I think that in order for this work to be considered for publication the analysis and conclusions put forward need to be substantiated by supporting modeling and theory. With this I do not recommend the manuscript for publication in its current form.

Reviewer #2 (Remarks to the Author):

The paper by Anwar et al. demonstrates the penetration of superconductivity (via differential conductance measurements) from the p-wave superconductor SRO214 through an interface into SRO113. Bulk p-wave superconductivity is extremely rare and SRO214 is the best known example, but so far all work has relied on bulk superconductivity measurements on single crystals due to the extreme sensitivity of p-wave pairing to electron scattering. The current work shows that it is possible to transfer superconductivity from single crystals of SRO214 into neighbouring magnetic materials which can be fabricated into devices - thus using the SRO214 as a substrate and source of p-wave pairs. This is an absolutely important achievement in its own right and following revision, the paper

should be published in Nature Communications. However, before publication the authors should revise the paper taking into account the following considerations:

Abstract

1. In the statement "Use of spin-triplet superconductors (TSCs) with active-spins" what does active mean. This is very unclear for - are the authors referring to spin-polarised spins?

Main text

2. Page 2, final paragraph. I do not see why using TSC means it is easier to control the LRPE. This seems like an unnecessarily statement which adds little to the justification of the paper as presumably a device exploiting spin-polarised Cooper pairs requires the control of spin downstream through magnetisation alignment.

3. Page 5 (end of). The authors claim the data in Fig. 2c is well fitted to the BCS gap equation. I do not agree - particularly for V1 and V2 as there is insufficient data to make this claim. Furthermore, this statement potentially weakens the paper as one of the potential exciting aspects of this result is that the p-wave components of SRO214 can leak into SRO113 through an interface which would not necessarily follow the BCS gap dependence on temperature. I advise the authors to remove this statement - in any case, density of states measurements are required in order to confirm the symmetry of the pairing wave function induced in SRO113. The short penetration length into SRO113 is also more consistent with p-wave pairing than s-wave odd frequency (due to the sensitivity of p-wave pairs to electron scatter).

4. Page 8. In addition to the mean free path for charge scatter, the second fundamental length scale that should limit the penetration length of spin-polarised Cooper pairs is the spin-flip length but the authors do not estimate a value for this. Can the authors please estimate this length and compare to the estimated coherence length.

5. A general remark: I would like the authors to emphasise in the introduction/conclusion why the transfer of p-wave superconductivity across an interface is important breakthrough as this is currently missing from the paper. Space could be saved by making fewer comparisons to the odd frequency proximity effects in superconductor / ferromagnet systems.

To summarise, this is an exciting result with robust data that will undoubtedly trigger interest from theorists working on superconductivity and experiments across the fields of superconductivity and spin-electronics.

[Hereafter, reviewers' comments are typed in blue color and authors' answers are in black color. Revisions of the manuscript are given in purple (revisions based on comments of Reviewer#1) and red color (Reviewer#2).]

Reviewers' comments and our responses:

Reviewer #1 (Remarks to the Author):

Report on: "Direct penetration of spin-triplet superconductivity into a ferromagnet in Au/SrRuO₃/Sr₂RuO₄ junctions" by M. S. Anwar et al.

In their manuscript the authors present an experiment that show evidence of a long-range proximity effect of the superconducting state of Sr₂RuO₄ in the ferromagnetic SrRuO₃. This proximity effect is probed by a conductance measurement using a Au electrode on top of the ferromagnet. Cooling the device down below 1.22 K two distinct reductions in resistance are seen and are correlated to: The superconducting transition in Sr₂RuO₄ at 1.22K as expected. The transition to a proximity superconductivity in the SrRuO₃ to layer. The SrRuO₃ layer is epitaxially grown on c-axis oriented Sr₂RuO₄ single crystal. This should give good electrical contact between the two materials.

The experimental findings in the low temperature regime of the conductance shows an emergence of an Andreev conductance at low bias. The bias window where the conductance is enhanced scales as the superconducting gap, either with temperature or with magnetic field, as expected from simple BCS theory. Furthermore, the conductance data shows a second structure found at higher biases with the same scaling as the low-bias feature. Combining the two features to a position-dependent proximity gap in SrRuO₃ a decay length, or effective coherence length, is extracted. This length exceeds estimates of a decay length expected in either a ballistic or diffuse ferromagnetic metal by factor between 3-10.

Q1. While I find the experiment very beautiful and the data extracted give a clear picture of a proximity effect extending over the 15nm SrRuO₃ ferromagnet, I find it hard to be convinced that this is an unequivocal evidence of Sr₂RuO₄ being a triplet superconductor. For this I would like to see some detailed modelling to support this claim.

A1. Thank you for evaluating our work as “very beautiful” and “give a clear picture of a proximity effect”. As we have already explained in the original manuscript, our experimental results, in particular, transmission electron microscopy of the interface, normal-state magnetoresistance, and conductance under applied field at temperature below T_c , provide strong evidence that we observed the long-range proximity effect without any interfacial magnetic inhomogeneity. A proximity effect without magnetic inhomogeneity is expected to occur only when spin-triplet correlation is induced directly from a spin-triplet superconductor, as displayed in Figures 1a and b.

Nevertheless, to strengthen our interpretation and to address the reviewer's concern, we performed calculations based on a recent theoretical model considering the Andreev reflection at a FM/TSC interface oriented along the c axis [D. Terrade *et al.*, PRB; Ref. 14 in the main text and Ref. S12 in supplementary information]. The configuration of this model is identical to the actual experimental configuration. We added a section "Theoretical model" in the Supplementary Information. For the FM, we considered an exchange field h_{ex} realistically corresponding to SrRuO₃. For the TSC, we assumed orbital symmetry p_x+ip_y , which is the most possible pairing symmetry for Sr₂RuO₄. We newly added a figure (Fig. S8) presenting the pair correlations with the Cooper-pair spin axis in the direction of the magnetization in the ferromagnet. This figure shows that spin-polarized Cooper pairs are indeed generated at the FM/TSC interface without any magnetic inhomogeneity and penetrated directly into the FM out of the TSC.

We comment here that the present manuscript is basically an experimental report. While the simple model gives a satisfactory result to explain our experimental observations, a more sophisticated theory is still needed. Such detailed theories are out of the scope of this experimental work and should be published elsewhere in future. We also believe that our work will stimulate theoretical investigations by other groups, as mentioned by Reviewer#2.

Q2. The chiral form of the pairing state in Sr₂RuO₄, the $(p_x + i p_y)$ orderparameter, would for instance average to zero in a c -axis oriented experiment as the current. How does this fact relate to the interpretation put forward?

A2. It is true that the $p_x + i p_y$ order parameter average to zero in case of the Josephson coupling along the c axis. This cancellation should result in absence of the critical current in c -axis oriented Josephson junctions. However, such a cancellation rule does not hold for the Andreev reflection, in which the momentum-resolved conductance does not cancel out on integration. This is already demonstrated by D. Terrade *et al.* [PRB **88**, 054516 (2013)], considering the c -axis oriented FM/TSC junction with the $p_x + i p_y$ order parameter in TSC. To address this point, we added following sentences into the main text of the revised manuscript,

"We should comment here on the chiral p -wave $p_x + i p_y$ order parameter of SRO214. In some types of experiments, superconducting contribution in transport along the c axis should vanish because the $p_x + i p_y$ order parameter is averaged to zero. For example, in c -axis oriented Josephson junctions between SRO214 and an s -wave superconductor, critical current is expected to be cancelled out to be zero. In contrast, the Andreev reflection in a metal attached along the c axis is certainly allowed even for the $p_x + i p_y$ state because the magnitude of the Andreev reflection is not given by the simple integration of the order-parameter phase¹⁴."

Q3. An other suggestion is that at the Sr₂RuO₄/SrRuO₃ interface there is a nucleation of a odd-frequency s -wave triplet pair amplitude. Such amplitude is always present where

reflection symmetry is broken. However, what is the theoretical signal in the low-bias conductance of such a pair-amplitude?

A3. Yes it is true that odd-frequency s -wave spin-triplet pairs can emerge at the SrRuO₃/Sr₂RuO₄ interface, when penetrated p -wave Cooper pairs are converted to s -wave pairs. This conversion occurs due to diffusive scatterings. Thus, for a diffusive system (system size $>$ mean free path) the odd-frequency s -wave pair amplitude will dominate, whereas for a clean system (system size $<$ mean free path) even-frequency p -wave amplitude will remain the majority. In our junctions, we used a 15-nm-thick SrRuO₃ layer, whose thickness is not longer than the electron mean free path (15-30 nm). Thus, it is anticipated that the dominant order parameter in our SrRuO₃ layer remains to be even-frequency p -wave spin-triplet correlation, as agreed by Reviewer #2. Theoretically, conductance spectra due to the odd-frequency pairs have been investigated but for non-magnetic junctions [Tanaka PRB 2005]. It has been revealed that diffusiveness is an important and required ingredient to achieve odd-frequency s -wave pair amplitude in a DN(diffusive-normal-metal)/TSC system. These theories predict that the odd-frequency correlation will generate the “mid-gap Andreev resonant state”, resulting in a sharp zero-bias peak appearing in the conductance. We expect that similar sharp zero-bias peak in conductance emerges even for FM/TSC systems, if odd-frequency pairs were dominant.

Experimentally, we did not observe any sharp zero bias anomaly. We observed rather flat-top conductance around the zero voltage. This fact, as well as the consideration of the size of our junction, supports that the even-frequency p -wave pair amplitude is dominant in our case. Of course, we never argue that the odd-frequency s -wave pair amplitude is totally absent. Some fraction of odd-frequency pairs must be there but out of observation.

To address this issue, we have added following sentences in the main text.

“Theoretically, it has been demonstrated that, for diffusive N/TSC junction, the odd-frequency s -wave correlation can emerge at the interface and lead to a sharp zero-bias peak in the conductance spectrum originating from the induced mid-gap Andreev resonant state³⁴. We anticipate that a similar zero-bias peak would be observed also in FM/TSC junctions if the odd-frequency pairs were dominant. On the other hand, the conductance of the present junctions at lower temperatures and zero field (see Fig. 2d) exhibits a flat-top peak around zero bias voltage, without any sharp anomalies. These observations also support the penetration of the p -wave (even-frequency) amplitude.”

I think that in order for this work to be considered for publication the analysis and conclusions put forward need to be substantiated by supporting modeling and theory. With this I do not recommend the manuscript for publication in its current form.

Appreciating his/her critical and constructive comments, we have added supporting model and theory, as we already explained above. By using an established model capturing a realistic experimental geometry, we clarified that the spin-polarized Cooper pair penetration into FM layer indeed occurs in our c -axis oriented heterostructure. This theoretical result strongly supports our conclusion of the direct and long-range spin-triplet

proximity effect. We believe that our responses and revisions are satisfactory for Reviewer #1 and make our manuscript publishable in Nature Communications.

Reviewer #2 (Remarks to the Author):

The paper by Anwar et al. demonstrates the penetration of superconductivity (via differential conductance measurements) from the p-wave superconductor SRO214 through an interface into SRO113. Bulk p-wave superconductivity is extremely rare and SRO214 is the best known example, but so far all work has relied on bulk superconductivity measurements on single crystals due to the extreme sensitivity of p-wave pairing to electron scattering. The current work shows that it is possible to transfer superconductivity from single crystals of SRO214 into neighbouring magnetic materials which can be fabricated into devices - thus using the SRO214 as a substrate and source of p-wave pairs. This is an absolutely important achievement in its own right and following revision, the paper should be published in Nature Communications. However, before publication the authors should revise the paper taking into account the following considerations:

Abstract

1. In the statement "Use of spin-triplet superconductors (TSCs) with active-spins" what does active mean. This is very unclear for - are the authors referring to spin-polarised spins?

A1. Here, we intended to refer to the spin-degree of freedom of spin-triplet Cooper pairs, which are not spin-polarized in bulk Sr_2RuO_4 at zero field, but possesses spin polarizability. To avoid the confusion and follow the suggestion of the reviewer, we have changed the phrase "*active spin*" to "*spin-polarizable Cooper pairs*".

Main text

2. Page 2, final paragraph. I do not see why using TSC means it is easier to control the LRPE. This seems like an unnecessarily statement which adds little to the justification of the paper as presumably a device exploiting spin-polarised Cooper pairs requires the control of spin downstream through magnetisation alignment.

A2. We have removed the words "*and controlling*" from the first line of the final paragraph at page 2. Indeed, to utilize this device, we need to align the spins of Cooper pairs with the magnetization of the ferromagnetic layer.

3. Page 5 (end of). The authors claim the data in Fig. 2c is well fitted to the BCS gap equation. I do not agree - particularly for V1 and V2 as there is insufficient data to make this claim. Furthermore, this statement potentially weakens the paper as one of the potential exciting aspects of this result is that the p-wave components of SRO214 can leak into SRO113 through an interface which would not necessarily follow the BCS gap dependence

on temperature. I advise the authors to remove this statement - in any case, density of states measurements are required in order to confirm the symmetry of the pairing wave function induced in SRO113. The short penetration length into SRO113 is also more consistent with p-wave pairing than s-wave odd frequency (due to the sensitivity of p-wave pairs to electron scatter).

A3. Following the suggestion of the reviewer, we have removed the BCS fitting results and also removed the statement *“Interestingly, the temperature dependence of V_2 and V_3 , as shown in Fig. 2c, is described well by the Bardeen-Cooper-Schrieffer (BCS) relationship for a superconducting gap. This fact supports our interpretation that V_2 and V_3 represent superconducting-gap-like features.”* Instead, we fit the $V_3(T)$ data with a theoretical temperature dependence of the superconducting gap of Sr_2RuO_4 obtained by taking into account the p -wave pairing, multi-orbital nature, and two dimensionality [Nomura, T., and Yamada, K., JPSJ **71**, 404 – 407 (2002). Ref. 29 of the revised manuscript]. The fitting was reasonably good, as shown in Fig. 2c of the revised manuscript. We added following statement *“The temperature dependence of characteristic voltages are shown in the Fig. 2c. Interestingly, $V_3(T)$ is described well with the temperature dependence of the superconducting gap of Sr_2RuO_4 obtained by taking into account the p -wave pairing and two dimensionality²⁹. This fact supports our interpretation that V_3 represent superconducting-gap-like features.”*

We agree with the reviewer that density-of-states measurements using a tunnel junction (N/I/F/T) or scanning tunneling spectroscopy are important to verify the symmetry of the Cooper pairs induced into SrRuO_3 . Such studies will be a wonderful future extension of our work.

4. Page 8. In addition to the mean free path for charge scatter, the second fundamental length scale that should limit the penetration length of spin-polarised Cooper pairs is the spin-flip length but the authors do not estimate a value for this. Can the authors please estimate this length and compare to the estimated coherence length.

A4. As the reviewer pointed out, the spin-flip length is an important parameter for the spin-triplet long-range proximity effect. We estimated the spin-flip length for SrRuO_3 thin films to be around 30 nm using an experimentally measured value of spin-flip time about 300 fs [Kanter, *et al.*, PRB 2011]. We have added the following paragraph just before the summary of the paper to evaluate the spin-flip length and compare it with other characteristic lengths.

“The LRPE can be also sensitive to spin-flip scattering events. Therefore, the Cooper pair penetration is also limited by the spin-flip length $L_{\text{sf}} = \sqrt{D_F \tau_{\text{sf}}}$, where τ_{sf} is the spin-flip time. Both s -wave (odd frequency) and p -wave (even frequency) correlation cannot extend over this length, whereas the latter can also be limited by the l_e . For SRO113 thin films, τ_{sf} is approximately ≈ 300 fs as revealed by an optical Kerr effect experiment³⁵, yielding $L_{\text{sf}} \approx 30$ nm. This estimated L_{sf} is three times larger than the value of the decay length $\xi_{113}^ \sim 10$*

nm obtained out from experiment. This comparison suggests that the proximity effect arising in our devices decays due to potential scatterings rather than spin-flip scatterings, in consistent with the scenario for the penetration of p -wave correlations.”

5. A general remark: I would like the authors to emphasise in the introduction/conclusion why the transfer of p -wave superconductivity across an interface is important breakthrough as this is currently missing from the paper. Space could be save by making fewer comparisons to the odd frequency proximity effects in superconductor / ferromagnet systems.

A5. We thank the reviewer for the suggestion. We have added the following sentences emphasizing the importance of p -wave Cooper pair transfer in the main text of our revised manuscript.

In introduction:

“In case of FM/TSC junctions, the orbital symmetry of the induced superconducting correlation is also important. In any superconductor-based junctions, because of the lack of inversion symmetry near the interface, pair amplitudes with various orbital symmetries (s -wave, p -wave, d -wave, ...) can in principle emerge at the interface¹³. Regardless the nature of the bulk superconductor, for a diffusive and clean junction the s -wave odd frequency and p -wave even frequency Cooper pairs dominate, respectively. So far, experimentally, s -wave odd frequency correlation has been observed in Josephson junctions (in diffusive limit) fabricated using various ferromagnets and s -wave SSCs. On the other hand, the p -wave correlation has never been realized. Thus, FM/TSC junctions provide opportunity to study properties of the p -wave induced pair amplitude¹⁴, in particular toward utilization of the orbital degree of freedom in combination with the spins.”

In summary:

“This observation opens new possibilities to transfer p -wave superconductor across an interface without losing spin information. Thus, our research offers new directions for the utilization of LRPE in FM/TSC systems in a new research area termed “*Superspintronics*.”

To summarise, this is an exciting result with robust data that will undoubtedly trigger interest from theorists working on superconductivity and experiments across the fields of superconductivity and spin-electronics.

We thank the reviewer for high evaluation of our work. With the revisions mentioned above, we believe that the manuscript is now acceptable for publication in Nature Communications.

REVIEWERS' COMMENTS:

Reviewer #1 (Remarks to the Author):

After reading the revised manuscript and taking part of the authors comments I´m even more convinced that the work needs to be substantiated by theoretical considerations. There is ample literature on proximity effect in SN and SF systems that could rather straight forwardly be applied in order to strengthen the claims the made base on the experiments. With out this backing, I feel that the results are rather open for several optional explanations.

So, I withstand my earlier opinion that the very nice experimental results need to be supported by theory in order to be accepted for publication in a high ranking journal.

Reviewer #2 (Remarks to the Author):

The authors have provided robust replies to all of my queries and have updated their paper where necessary. As I stressed in my first report, this paper provides important new insight into the proximity coupling of superconductivity in SRO through a magnetic (SRO 113) interface into a non-magnetic, non-superconducting, material (Au). This is an important step in the field as the results could lead to the development of devices based on p-wave superconductivity, which is virtually impossible to achieve using isolated single crystal crystal SRO.

Response to referees

Reviewer #1 (Remarks to the Author):

After reading the revised manuscript and taking part of the authors comments I´m even more convinced that the work needs to be substantiated by theoretical considerations. There is ample literature on proximity effect in SN and SF systems that could rather straight forwardly be applied in order to strengthen the claims the made base on the experiments. With out this backing, I feel that the results are rather open for several optional explanations.

So, I withstand my earlier opinion that the very nice experimental results need to be supported by theory in order to be accepted for publication in a high ranking journal.

Our response:

We are pleased that this reviewer appreciates our experimental achievement as “the very nice experimental results”. But he insists us to explain our work with rather systematic and straightforward theory. We agree that a straightforward and detailed theory is in future necessary to understand our results. However, as we explained in the previous round, there are only a few theories that can properly model our junction. For a meaningful comparison of our results with theory, we cannot use existing theories on SN or SF systems but we have to fully construct a TFN model with a relevant quasi-two-dimensional electronic structure. Such substantial theoretical effort is out of the scope for this present manuscript, reporting the first experimental observation of the penetration of superconductivity from a spin-triplet superconductor to a ferromagnet. We believe that our work stimulate extensive theoretical studies to understand the spin-triplet proximity effect between a ferromagnet and a spin-triplet superconductor.

Reviewer #2 (Remarks to the Author):

The authors have provided robust replies to all of my queries and have updated their paper where necessary. As I stressed in my first report, this paper provides important new insight into the proximity coupling of superconductivity in SRO through a magnetic (SRO 113) interface into a non-magnetic, non-superconducting, material (Au). This is an important step in the field as the results could lead to the development of devices based on p-wave superconductivity, which is virtually impossible to achieve using isolated single crystal crystal SRO.

Our response:

We are thankful to this reviewer for appreciating our work. We also appreciate his expert analysis to understand the importance of our work.